

# The effect of supragingival glycine air polishing on periodontitis during maintenance therapy: a randomized controlled trial

Hongye Lu[1,2], Lu He[1,2], Yibing Zhao[1,2] and Huanxin Meng[1,2]

[1] Department of Periodontology, Peking University School and Hospital of Stomatology, Beijing, China
[2] National Engineering Laboratory for Digital and Material Technology of Stomatology, Beijing Key Laboratory of Digital Stomatology, Beijing, China

Corresponding authors
Yibing Zhao, zhaoyb2000@sina.com
Huanxin Meng, kqhxmeng@bjmu.edu.cn

## ABSTRACT

**Background:** Glycine air polishing has been proved to be safe, comfortable and time-saving. Whether it could substitute ultrasonic scaling to remove dental plaque biofilm during periodontal maintenance remains unclear. The purposes of this study were to evaluate the effect of supragingival glycine air polishing (SGAP) on the subgingival periodontal pathogens during maintenance therapy and to check the association of periodontal pathogens and clinical parameters.

**Methods:** Twenty-three chronic periodontitis patients during their maintenance therapy were enrolled in the 12-week study. According to randomized split-mouth design, the test side was treated with SGAP (65 $\mu$m), while the control side was treated with supragingival ultrasonic scaling and polishing (SUSP) with rubber cup. Clinical examination including plaque index (PLI), probing depth (PD), bleeding index (BI) were performed at baseline and 12 weeks post-treatment. Sampling of the subgingival plaque at each investigational site (mesiobuccal site of the mandibular first molar) was performed at baseline and 2, 4, 8, 12 weeks after maintenance treatment. Four periodontal pathogens including *Porphyromonas gingivalis*, *Tannerella forsythia*, *Treponema denticola* and *Fusobacterium nucleatum* were detected by 16S rDNA polymerase chain reaction.

**Results:** Clinical status generally improved after treatment in both groups. PLI in both groups, PD in SGAP group and bleeding on probing (%) in SUSP group significantly decreased after treatment ($p < 0.05$). There was no significant difference of clinical parameters between two groups before and after treatment. The detection rates of *P. gingivalis*, *T. denticola* in both groups, *T. forsythia* in SUSP group and *F. nucleatum* in SGAP group decreased after maintenance treatment in both groups, although no significant difference was found, and it rebound to baseline level at 12 weeks after maintenance treatment. There was no significant difference between SGAP group and SUSP group at any time point. *T. denticola*-positive sites had significantly greater BI than *T. denticola*-negative sites ($p < 0.05$).

**Discussion:** Supragingival glycine air polishing had a reliable effect in removing subgingival dental plaque biofilm during maintenance period, and three months may be a proper maintenance interval for pockets not more than 5 mm.

# INTRODUCTION

Periodontitis is the leading factor of tooth loss for adults (*Murray et al., 1997*; *Phipps & Stevens, 1995*), and it is initiated by dental plaque biofilm (*Gold, 1985*; *Witzel, 1882*). Therefore, removal of periodontal plaque is of great importance and necessity (*Matuliene et al., 2008*; *Rosling et al., 2001*). As patients' self-performed oral hygiene is inadequate to remove the newly formed biofilm, patients with periodontitis who have completed their active periodontal treatment should receive supportive periodontal treatment repeatedly (*Axelsson & Lindhe, 1981*; *Axelsson, Nystrom & Lindhe, 2004*). Scaling and root planing by hand or ultrasonic instruments is generally adopted to remove biofilm during the maintenance phase (*Lindhe & Nyman, 1984*). However, it is both time-consuming and discomfortable, and it may lead to the damage of dentin and cementum to some extent (*Petersilka et al., 2003b*; *Zappa et al., 1991*). Air polishing has been accepted to be an effective alternative to remove biofilm and stain (*Petersilka et al., 2003c*; *Zhao, He & Meng, 2015*), which makes maintenance therapy more time-saving and comfortable. The original material used in air polishing was sodium bicarbonate (*Weaks et al., 1984*; *Willmann, Norling & Johnson, 1980*), which could cause severe gingival erosion and substantial root damage (*Petersilka et al., 2003a*; *Simon, Munivenkatappa Lakshmaiah Venkatesh & Chickanna, 2015*). Comparing with sodium bicarbonate, glycine is odorless, low-abrasive and highly water-soluble (*Bozbay et al., 2016*). Furthermore, glycine has been proved to have immunomodulatory, anti-inflammatory and cytoprotective effect for periodontal tissue (*Breivik et al., 2005*; *Schaumann et al., 2013*), which makes it an excellent material for periodontal air polishing. Previous study demonstrated that glycine air polishing could save half the treatment time and double the comfort comparing with ultrasonic scaling and polishing with rubber-cup (*Zhao, He & Meng, 2015*).

Two main types glycine air polishing are used in the periodontal treatment. One type is supragingival glycine air polishing (SGAP) with standard air polishing nozzle and 65 $\mu$m glycine powder; the other type is subgingival glycine air polishing with subgingival nozzle and 25 $\mu$m glycine powder (*Cobb et al., 2017*). Several studies have evaluated the clinical and microbiological efficiency of subgingival glycine air polishing. A short-term study compared the effect of subgingival air polishing on 5–8 mm periodontal pockets in 20 patients with ultrasonic scaling during supportive periodontal therapy (SPT). Bleeding on probing (BOP), probing depth (PD), attachment loss (AL) and bacteria decreased after treatment, but no significant difference between air polishing and ultrasonic scaling was found (*Wennstrom, Dahlen & Ramberg, 2011*). *Flemmig et al. (2012)* conducted a 90-day study on 30 patients with chronic periodontitis after initial therapy, they applied subgingival air polishing in 4–9 mm pockets, and significant reduction was observed in the count of *Porphyromonas gingivalis* and *Tannerella forsythia* in moderate-deep pockets. A 12-month randomized controlled trial had evaluated the long-term effect of subgingival air polishing. The number of pockets >4 mm, PD and

BOP significantly reduced at month 12 after treatment. And the prevalence of *Aggregatibacter actinomycetemcomitans* reduced more in air polishing group than ultrasonic debridement group (*Muller et al., 2014*). The short-term and long-term efficacy of subgingival air polishing on moderate-to-deep pockets had been confirmed in different studies.

Periodontitis during maintenance period should have inflammation well controlled with shallow pockets not more than 5 mm (*Matuliene et al., 2008*). Nevertheless, few studies about glycine air polishing focused on this area. *Flemmig et al. (2007)* applied standard air polishing unit in hopeless teeth, then extracted and stained them to assess the microbiological effect of supragingival air polishing. The morphological result revealed that air polishing debridement efficacy extended up to 5.17 mm in the subgingival surfaces, and in sites with PD of 3–5 mm, the debridement depth reached apically 65–45% of the periodontal pockets. It suggested that SGAP could be used to remove biofilm in pockets not more than 5 mm during maintenance period. However, very few studies about air polishing involved in this area. *Petersilka et al. (2003c, 2003d)* applied supragingival air polishing in 3–5 mm periodontal pockets and evaluated the subgingival microbiological effect of supragingival air polishing by anaerobe cultivation. Subgingival bacterial counts reduced dramatically by >99% after air polishing and debridement with air polishing was superior to that with curettes. Although previous studies showed satisfying effect of air polishing immediately after treatment, they did not answer how long the microbiological effect would last after glycine air polishing, neither did they evaluate the clinical effect.

It is assumed that supragingival air polishing could effectively remove periodontal pathogens in pockets not more than 5 mm during maintenance therapy and that the change of periodontal pathogens was in accordance with clinical change. Thus, the objective of the study was to evaluate the follow-up clinical and microbiological effect of SGAP comparing with ultrasonic scaling and polishing during maintenance phase and to study the relation of periodontal pathogens and clinical changes.

## MATERIALS AND METHODS

This was an examiner-masked randomized controlled clinical trial with 12-week duration. The split-mouth method was adopted to compare the effect of SGAP with supragingival ultrasonic scaling and polishing (SUSP) with rubber-cup. This study protocol was approved by Biomedical Ethics Committee of the Peking University School and Hospital of Stomatology (approval number: IRB00001052-05106). The clinical study was registered at Chinese Clinical Trial Registry, and the approval number was ChiCTR-INR-17013073. All participating patients were given information about the study and signed informed consent before the inclusion.

### Sample size calculation

According to the microbiological change during maintenance therapy of previous study (*Wennstrom, Dahlen & Ramberg, 2011*), *P. gingivalis* was the primary outcome and

maximum allowable difference was 20%. Non-inferiority test for two correlated proportions was performed with significant level $\alpha = 0.05$ and $\beta = 0.2$, and the minimum size was calculated as 20 using software PASS version 11 (NCSS, USA). Considering the rate of possible loss to follow-up 15%, sample size was set as 23.

## Participants

Twenty-three unrelated Chinese patients with chronic periodontitis were recruited in the study. They all had completed their active treatment and had been maintained for several years at the Department of Periodontology, Peking University School and Hospital of Stomatology, China between August 2011 and March 2013.

Inclusion criteria were as follows:

- Patients with periodontitis had completed comprehensive periodontal therapy at least three months and were in their maintenance period.
- Periodontal PD not more than 5 mm.
- Remaining more than 20 teeth.
- Without systemic disease.
- No smoking.

Exclusion criteria were as follows:

- Pregnancy or lactation.
- Scaling and root planing within three months.
- Having periodontal surgery within six months.
- Antibiotic therapy within three months.

## Randomization and interventions

This trial was designed as a randomized split-mouth study. Subject numbers were assigned at the enrollment visit in ascending order. RV.UNIFORM (0,1) was used to generate the randomization table with SPSS v.20.0 (IBM Corp, Armonk, NY, USA). Then each subject got a random number. According to random number of each subject, the right half-mouth and the left half-mouth were divided into test or control group, respectively. The test group was conducted with SGAP; while the control group was conducted with SUSP with rubber-cup and paste. Curettes were used to remove the remaining hard calculus if necessary. The intervention was performed by an experienced senior periodontist. Any discomfort or pain during or after treatment should be recorded. The flowchart of the study was showed in Fig. 1.

Supragingival air polishing was conducted with 65 $\mu$m amino acid glycine powder (Air-Flow Polishing Soft; EMS, Nyon, Switzerland), Air-Flow handy2 (EMS, Nyon, Switzerland) and Air-Flow Masters (EMS, Nyon, Switzerland). Ultrasonic scaling was conducted with the use of ultrasonic instrument (Satelec, Merignac, France), contra-angle handpiece (NSK, Tochigi, Japan), rubber-cup and polishing paste (Ivoclar, Schaan, Liechtenstein).

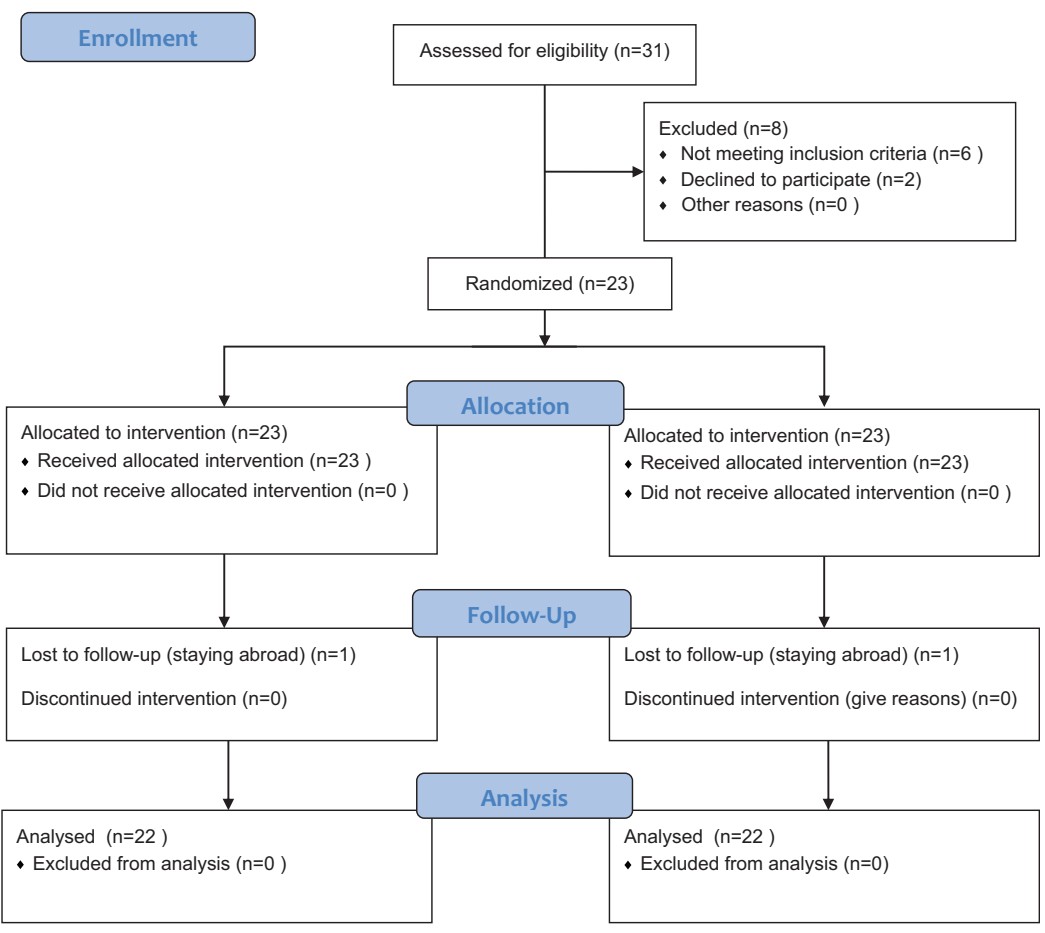

**Figure 1  Flow diagram of the progress through the phases of the randomized trial.** SGAP group, supragingival glycine air polishing group; SUSP group, supragingival ultrasonic scaling group.

## Clinical examination

Clinical periodontal examination was performed at baseline before treatment and the endpoint at 12 weeks. It was carried out by an experienced periodontist, whose self-consistency test showed substantial consistency, with 96% of the sites differing 1 mm or less for pocket depth measurements. The examiner was masked with the general information and treatment information of recruited subjects. Criteria for periodontal clinical examination were as follows:

(1) Plaque index (PLI): *Silness & Loe (1964)*.
(2) Probing depth: The distance from the bottom of periodontal pockets to gingival margin using Williams probe.
(3) Bleeding index (BI): *Mazza, Newman & Sims (1981)*.
(4) Bleeding on probing: The sites with BI $\geq 2$ were recorded as BOP-positive.

Participants received oral hygiene instruction after clinical examination every time.

## Sample collection

Samples of the subgingival microbiota at each investigational site was collected at baseline and 2, 4, 8 and 12 weeks after treatment. Isolated the sampled area, removed the supragingival dental plaque, dried the supragingival tooth surface and collected subgingival dental plaque with sterile curettes in the mesiobuccal sites of two first molars in mandible (if the first molar was lost, the second molar in mandibular would be investigated), and then transferred the samples into sterile Eppendorf tube and stored them in −80 °C. Sampling was performed before clinical examination to avoid interacting infection and destruction of plaque by periodontal probing.

## DNA extraction and polymerase chain reaction measurements

The genomic DNA of bacteria was extracted from subgingival dental plaque. Wash the plaque with 500 μl TE buffer (10 mM Tris–HCl, 1 mM EDTA, pH 7.6) for three times. Add 180 μl 20 mg/ml lysozyme (Solarbio, Beijing, China) to the deposit, and water bath at 37 °C overnight. Microscale genomic DNA extraction kit (TIANGEN, Beijing, China) was used, according to the manufacturer's instructions. Finally, the concentration and purity of DNA were checked by Nanodrop 2000 (Thermo Scientific, Waltham, MA, USA).

The samples were analyzed for the detection of four periodontal pathogenic bacteria including *P. gingivalis*, *T. forsythia*, *T. denticola*, *F. nucleatum* using 16S rRNA-based polymerase chain reaction (PCR). The DNA extracted from *P. gingivalis* (ATCC 51700), *T. forsythia* (ATCC 43037), *T. denticola* (ATCC 33520) and *F. nucleatum* (ATCC 25586) were used as positive control for PCR analysis, and ultrapure water was used as negative control. The definitive primer followed the reported research of *Ashimoto et al. (1996)* and *Baumgartner et al. (2004)* showed in Table 1.

Polymerase chain reaction amplification was carried out with GeneAmp PCR system 2700 (ABI, South San Francisco, CA, USA), and the parameters were as follows: PCR mixtures contained 2 μl template, 1 μl primer (10 μM) primer, 2.5 μl 10× buffer $Mg^{2+}$ plus, 2 μl dNTP (2.5 mM), 0.2 μl Taq DNA polymerase (5 U/μl) (TaKaRa Biotechnology, Dalian, China), 17.3 μl $ddH_2O$. Thermocycling for *P. gingivalis*, *T. forsythia*, *T. denticola* was 95 °C 2 min, 95 °C 30 s, 60 °C 1 min, 72 °C 1 min, 36 cycles; 72 °C 2 min. Thermocycling for *F. nucleatum* was 94 °C 1 min, 60 °C 1 min, 72 °C 90 s, 35 cycles; 72 °C 10 min, 95 °C 2 min, 95 °C 30 s, 60 °C 1 min, 72 °C 1 min, 36 cycles; 72 °C 2 min.

Polymerase chain reaction products were analyzed by Horizontal Gel Electrophoresis with 1.5% agarose (Solarbio, Beijing, China), 130 V. DL 2000 ladder (TaKaRa Biotechnology, Dalian, China) served as the DNA weight marker. The agarose gel was stained with GoldView type I (Solarbio, Beijing, China) and taken radiogram with 300 nm ultraviolet illumination.

## Statistical analysis

The clinical parameters of SGAP group and SUSP were tested for normality with Kolmogorov–Smirnov method. The parameters following the normal distribution would be tested by paired Student's *t*-test to compare the difference of two groups.

**Table 1** Primers used in PCR analysis.

| Primer sequence (5′–3′) | | Base position (length) | Reference |
|---|---|---|---|
| *Porphyromonas gingivalis* | | | |
| Forward | AGG CAG CTT GCC ATA CTG CG | 729–1,132 (404) | *Ashimoto et al. (1996)* |
| Reverse | ACT GTT AGC AAC TAC CGA TGT | | |
| *Tannerella forsythia* | | | |
| Forward | GCG TAT GTA ACC TGC CCG CA | 120–760 (641) | *Ashimoto et al. (1996)* |
| Reverse | TGC TTC AGT GTC AGT TAT ACC T | | |
| *Treponema denticola* | | | |
| Forward | TAA TAC CGA ATG TGC TCA TTT ACA T | 193–508 (316) | *Ashimoto et al. (1996)* |
| Reverse | TCA AAG AAG CAT TCC CTC TTC TTC TTA | | |
| *Fusobacterium nucleatum* | | | |
| Forward | AGGGCATCCTAGAATTATG | 190–1,006 (817) | *Baumgartner et al. (2004)* |
| Reverse | GGGACACTGAAACATCTCTGTCTCA | | |

The paired non-parametric test should be applied with Wilcoxon method if variables did not follow normal distribution. Mann–Whitney test was used to compare independent samples which did not follow normal distribution. The clinical parameters of full mouth in SGAP group and SUSP group were accumulated at patient level. The frequency of investigated periodontal pathogens between SGAP group and SUSP group was analyzed by McNemar's test. Longitudinal data was analyzed by McNemar's test for pairwise comparisons. The Spearman rank correlation coefficient was used to determine the rank correlation of clinical parameters and periodontal pathogens. The significance level was preset at 0.05 for all statistical tests. SPSS was used to carry out the statistical analyses.

## RESULTS

### Subjects

Twenty-two patients, consent to the information, were recruited in the study. They all had initially moderate to severe chronic periodontitis, and they all had regular periodontal maintenance therapy before enrollment. There were 8 males and 14 females, with their age ranging from 28 to 72 years old. And one patient lost to follow-up at week 4 for staying abroad. Totally, 11 subjects' left sides and 12 subjects' right sides were assigned to test group, and all patients were right handed. In all, 290 teeth, 1,740 sites in SGAP group and 291 teeth, 1,746 sites in SUSP group were clinically examined and 222 samples (111 in each group) were collected and detected. There was no adverse event during the study period.

### Clinical assessment

Clinical parameters of full mouth at individual level in SGAP group and SUSP group were presented in Table 2. The clinical parameters at baseline in SGAP group and SUSP group were undifferentiated, and there were no differences at week 12. All of the clinical parameters declined at week 12 after maintenance treatment in both groups.

**Table 2 Clinical parameters of all investigational sites in SGAP group and SUSP group.**

|  | SGAP group ($n$ = 22) | SUSP group ($n$ = 22) | $p$ Value |
|---|---|---|---|
| *PLI* | | | |
| Baseline | 0.54 ± 0.25[*] | 0.59 ± 0.33[*] | 0.267 |
| 12 weeks | 0.46 ± 0.34[*] | 0.44 ± 0.24[*] | 0.733 |
| Reduction | 0.08 ± 0.33 | 0.15 ± 0.17 | 0.230 |
| $p$ Value | 0.046 | 0.000 | |
| *PD (mm)* | | | |
| Baseline | 2.48 ± 0.32[*] | 2.47 ± 0.30 | 0.693 |
| 12 weeks | 2.34 ± 0.32[*] | 2.38 ± 0.27 | 0.291 |
| Reduction | 0.14 ± 0.23 | 0.09 ± 0.23 | 0.091 |
| $p$ Value | 0.013 | 0.080 | |
| *BI* | | | |
| Baseline | 1.05 ± 0.31 | 1.08 ± 0.37 | 0.547 |
| 12 weeks | 0.96 ± 0.29 | 0.97 ± 0.26 | 0.961 |
| Reduction | 0.11 ± 0.36 | 0.11 ± 0.41 | 0.745 |
| $p$ Value | 0.249 | 0.230 | |
| *BOP (%)* | | | |
| Baseline | 21.82 ± 11.76 | 24.14 ± 13.32[*] | 0.368 |
| 12 weeks | 18.01 ± 11.38 | 17.50 ± 10.35[*] | 0.315 |
| Reduction | 3.81 ± 10.6 | 6.64 ± 12.04 | 0.334 |
| $p$ Value | 0.107 | 0.017 | |

Notes:
SGAP group, supragingival glycine air polishing group; SUSP group, supragingival ultrasonic scaling group; PLI, plaque index; PD, probing depth; BI, bleeding index; BOP, bleeding on probing.
[*] Significant difference between baseline and 12 weeks.

PLI dramatically dropped from 0.54 and 0.59 to 0.46 and 0.44 in SGAP group and SUSP group, respectively. And significant difference was observed ($p$ < 0.05). PD in SGAP group significantly decreased from 2.48 to 2.34 mm ($p$ < 0.05), and the reduction was 0.14 mm. PD in SUSP group decreased from 2.47 to 2.38 mm and the reduction was 0.09 mm, although no significant difference was found ($p$ = 0.080). The BI slightly declined after treatment (SGAP group: from 1.05 to 0.96; SUSP group: from 1.08 to 0.97). The percentage of BOP reduced after treatment (SGAP group: from 21.82% to 18.01%; SUSP group: from 24.14% to 17.50%), and the change between baseline and week 12 in SUSP group show significant difference ($p$ < 0.05).

Table 3 displayed the clinical parameters of sampling sites in SGAP group and SUSP group. At baseline, all parameters were homogenous. Sampling sites had mean PD of 2.90 mm in SGAP group and 3.00 mm in SUSP group, respectively. At 12 weeks after treatment, the mean PD was 2.95 mm in SGAP group and 3.14 mm in SUSP group, and there was no site with PD more than 5 mm. No significant difference was found in PD between baseline and 12-week time point. Both groups showed relatively improved periodontal status. Mean BI in SGAP group and SUSP group was respectively 1.32 and 1.27 at baseline, and it became 1.29 and 1.14 at week 12. A reduction of BI at 12 weeks after treatment was observed in both groups, although no significant

**Table 3 Clinical parameters of sampling sites in SGAP group and SUSP group.**

| | SGAP group ($n = 22$) | SUSP group ($n = 22$) | $p$ Value |
|---|---|---|---|
| *PLI* | | | |
| Baseline | 0.36 ± 0.49 | 0.54 ± 0.51* | 0.366 |
| 12 weeks | 0.50 ± 0.74 | 0.27 ± 0.16* | 0.194 |
| $p$ Value | 0.439 | 0.014 | |
| *PD (mm)* | | | |
| Baseline | 2.91 ± 0.87 | 3.00 ± 0.87 | 0.917 |
| 12 weeks | 2.95 ± 0.90 | 3.14 ± 0.47 | 0.506 |
| $p$ Value | 0.862 | 0.334 | |
| *BI* | | | |
| Baseline | 1.32 ± 0.84 | 1.27 ± 0.70 | 0.854 |
| 12 weeks | 1.23 ± 0.69 | 1.09 ± 0.81 | 0.536 |
| $p$ Value | 0.672 | 0.248 | |

**Notes:**
SGAP group, supragingival glycine air polishing group; SUSP group, supragingival ultrasonic scaling group; PLI, plaque index; PD, probing depth; BI, bleeding index.
* Significant difference between baseline and 12 weeks.

difference was found. Supragingival dental plaque measured by PLI significantly decreased in SUSP group ($p < 0.05$). There was no significant difference between SGAP group and SUSP group in terms of PD, BI and PLI at baseline or 12 weeks.

## Microbiological assessments

The detection frequencies of periodontal pathogens in the present study were displayed in Fig. 2. Before treatment, the detection frequencies of *P. gingivalis*, *T. forsythia*, *T. denticola* and *F. nucleatum* between SGAP group and SUSP group had no significant differences. After treatment, a general trend of detection frequency reduction was found in both two groups. However, few significant differences were observed when comparing detection frequencies of different time points within groups. At week 12 the detection frequencies had returned to the level before treatment.

Detection frequencies of *P. gingivalis*, *T. denticola* and *F. nucleatum* in both groups declined at week 2 after treatment comparing with baseline (SGAP group: from 26.1% to 13.0%; SUSP group: from 34.8% to 13.0%), though no significant differences were observed. It kept a relatively low frequency throughout the observation period in SUSP group and rebound subsequently in SGAP group. Regarding *T. forsythia*, the prevalence did not change much at different time points in SGAP group; while in SUSP group, it decreased from 56.5% to 34.8% during the first two weeks after treatment, kept a low detection frequency and increased at 12 weeks. Detection frequencies of *T. denticola* dropped off after treatment and then increased at week 4 in both groups. More reduction in SUSP group was observed than that in SGAP group (SGAP group: from 39.1% to 26.1%; SUSP group: from 30.4% to 8.7%). The detection frequency of *F. nucleatum* did not show change before and after treatment in both groups. At any time point, the two groups didn't show any significant difference in the detection frequencies of all four periodontal pathogens.

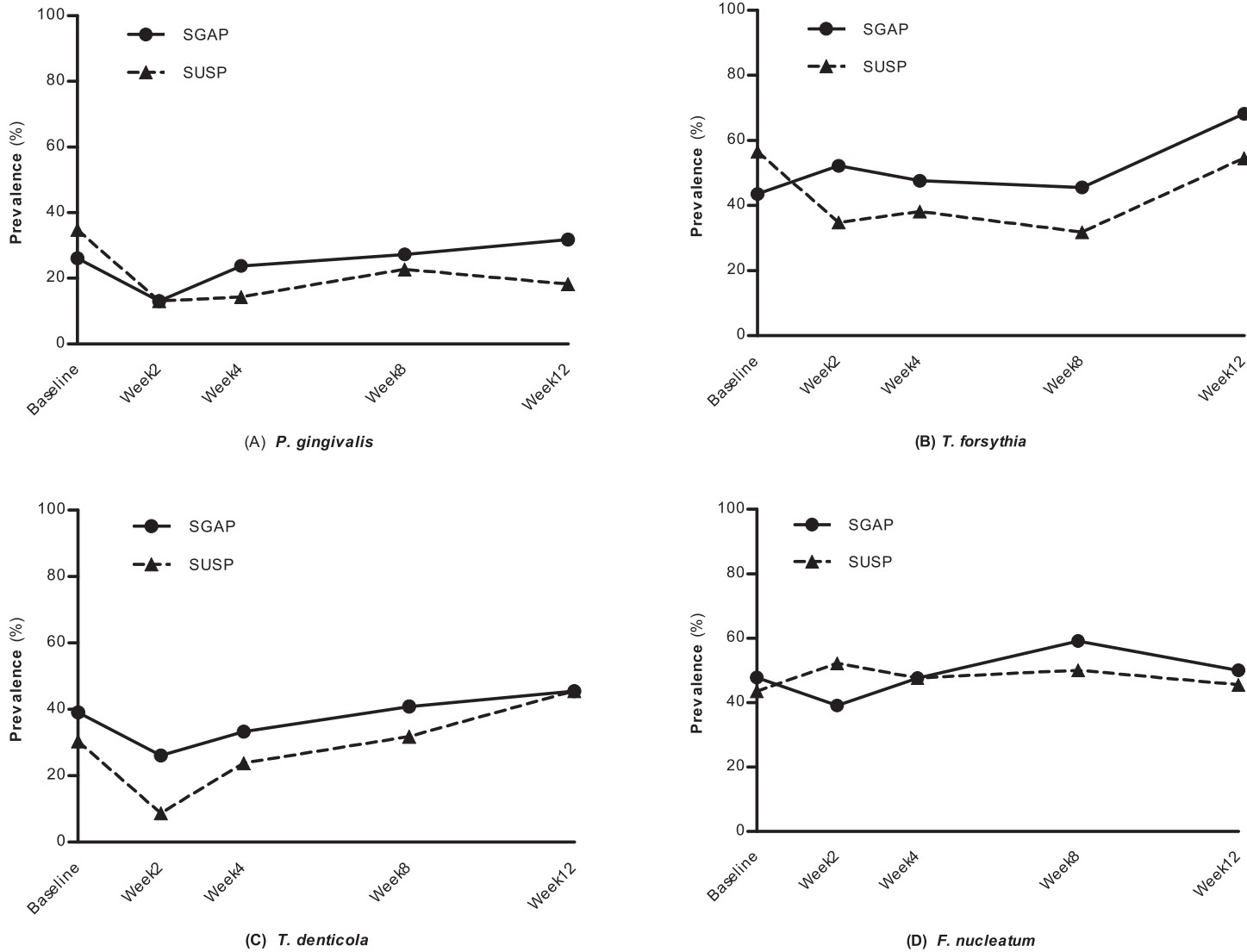

**Figure 2 The prevalence of four periodontal pathogens at different time points.** SGAP group, supragingival glycine air polishing group; SUSP group, supragingival ultrasonic scaling group. (A) *P. ginigvalis*, (B) *T. forsythia*, (C) *T. denticola*, (D) *F. nucleatum*.

The comparisons of PD and BI between periodontal pathogen-positive and -negative sites revealed that *T. denticola* exhibited significant difference with BI (Table 4). *T. denticola*-positive sites had significantly greater BI than *T. denticola*-negative sites ($p < 0.05$). Strong association between *T. denticola* and BI was observed ($p < 0.05$).

## DISCUSSION

The clinical status had improved after treatment, and the prevalence of periodontal pathogens in subgingival plaque generally decreased after being treated by supragingival air polishing or ultrasonic scaling and polishing. Previous study adopted SGAP in 3–5 mm pockets and found that subgingival bacterial counts significantly reduced immediately after air polishing (*Petersilka et al., 2003c*, *2003d*). Our study evaluated

**Table 4 Comparison of PD and BI between periodontal pathogens-positive and -negative sites.**

|  | PLI | PD (mm) | BI |
|---|---|---|---|
| *P. gingivalis* |  |  |  |
| Positive | 0.60 ± 0.71 | 2.96 ± 0.61 | 1.44 ± 0.71 |
| Negative | 0.35 ± 0.48 | 2.95 ± 0.93 | 1.17 ± 0.74 |
| *p* Value | 0.148 | 0.939 | 0.150 |
| *T. forsythia* |  |  |  |
| Positive | 0.46 ± 0.59 | 2.92 ± 0.83 | 1.32 ± 0.65 |
| Negative | 0.38 ± 0.54 | 3.00 ± 0.88 | 1.15 ± 0.83 |
| *p* Value | 0.484 | 0.586 | 0.286 |
| *T. denticola* |  |  |  |
| Positive | 0.39 ± 0.49 | 3.11 ± 0.82 | 1.47 ± 0.65* |
| Negative | 0.44 ± 0.60 | 2.85 ± 0.86 | 1.09 ± 0.76* |
| *p* Value | 0.839 | 0.107 | 0.030 |
| *F. nucleatum* |  |  |  |
| Positive | 0.43 ± 0.55 | 2.90 ± 0.69 | 1.28 ± 0.74 |
| Negative | 0.42 ± 0.58 | 3.00 ± 0.97 | 1.21 ± 0.74 |
| *p* Value | 0.835 | 0.558 | 0.803 |

Notes:
SGAP group, supragingival glycine air polishing group; SUSP group, supragingival ultrasonic scaling group; PLI, plaque index; PD, probing depth; BI, bleeding index.
* Significant difference between baseline and 12 weeks.

the microbiological effect of air polishing for longer term with a 12-week follow-up, and the result further confirmed the microbiological effect of SGAP at 2, 4 and 8 weeks after maintenance therapy. It indicated that SGAP could remove subgingival periodontal pathogen bacteria in ≤5 mm periodontal pockets effectively.

However, air polishing lacks the ability to remove well-mineralized deposit for its low abrasiveness. For patients with firmly attached calculus, ultrasonic scalers or hand instruments are still needed. All the subjects recruited in this study had regular maintenance therapy before enrollment, and only two of them had calculus. The calculus mainly existed in lingual aspect of mandibular anterior teeth, and it was removed by curettes with little effort, and it just cost seconds.

Prevalence of periodontal pathogens decreased after treatment and almost rebounded to the pre-treatment level at 12 weeks, which indicated that dental plaque biofilm could reconstruct within 12 weeks. Similar result was found with subgingival air polishing that total count of subgingival bacteria returned to baseline at 90 days after treatment (*Flemmig et al., 2012*). A randomized controlled trial evaluated the effect of different maintenance recall intervals in patients with chronic periodontitis treated by full-mouth ultrasonic debridement and drew the same conclusion that SPT at three-month intervals promoted short-term stability of clinical improvements (*Ueda et al., 2014*). The clinical and microbiological evidence suggested that the interval of air polishing and ultrasonic scaling could be three months during maintenance period. In addition, within this interval, dental plaque would not mineralize into calculus so that it can be removed by air polishing well (*White, 1997*).

In the present study, no significant differences in clinical parameters and prevalence of four periodontal pathogens between two approaches was observed at any time point. It indicated that SGAP was equivalent with ultrasonic scaling and polishing with rubber-cup in removing subgingival periodontal pathogens in PD ≤5 mm sites during maintenance therapy. The comparison between subgingival air polishing and subgingival ultrasonic scaling also drew the same conclusion. A 60-day clinical trial assessed the clinical and microbiological effect of subgingival glycine air polishing comparing subgingival ultrasonic scaling in moderate deep pockets, and it demonstrated no pertinent differences (*Wennstrom, Dahlen & Ramberg, 2011*). The immediate comparison before and after treatment described by *Petersilka et al. (2003c*, *2003d)* showed that low-abrasive air polishing was superior to curettes in removing subgingival plaque at sites with up to 5 mm PD during maintenance period. Because they adopted sterile paper point to sample, which mainly took unattached plaque. Whereas, scaling and root planing by curettes mainly removed attached plaque. Therefore, their results showed less reduction of bacteria after being treated with curettes.

In this study, the prevalence of *P. gingivalis* ranged from 13.0% to 31.8% in SGAP group and from 13.0% to 34.8% in SUSP group during the observational period. A study in Chinese population showed that the prevalence of *P. gingivalis* in untreated periodontitis was as high as 93.0% (*Ding et al., 2010*). The prevalence in our study was much lower than that in untreated periodontitis. As the keystone of periodontal pathogens, low prevalence of *P. gingivalis* indicated good infection control (*Hajishengallis, Darveau & Curtis, 2012*). A study detected *P. gingivalis* at 3, 6, 9 and 12 months after active periodontal therapy, and the prevalence ranged from 8% to 15% (*Cugini et al., 2000*). To address this discrepancy, that participants had initially moderate to severe periodontitis before periodontal treatment should be considered.

*Treponema denticola* was suspected as a major periodontal pathogen which associated with the severity of periodontitis (*Takeuchi et al., 2001*). Our result demonstrated that *T. denticola* was significantly associated with BI, which was the most sensitive parameter of periodontal inflammation. Volatile fatty acids, such as succinic acid, acetic acid, propionic acid, butyric acid and isovaleric acid, were found higher in *T. denticola*-positive sites than negative sites in aggressive periodontitis patients (*Lu et al., 2013*). Previous study had also proved that PLI, GI (gingiva index), PD, AL were significantly higher in *T. denticola*-positive sites than negative sites (*Yuan et al., 2001*). The result demonstrated *T. denticola* has close relationship with inflammation.

The result of present study displayed that the prevalence of *F. nucleatum* did not change among different time points and persistently maintain a relatively high prevalence. *F. nucleatum*, as an intermediate colonizer bridging, is one of the initial gram-negative species established in biofilm (*Kolenbrander, 2000*; *Kolenbrander et al., 2002*). It could restore to the level before treatment within very short time and exist in subgingival biofilm as a prominent component in quantity.

In the study, the split-mouth method avoided the interference of complex confounding factors from individual difference. One side usually has similar periodontal status to the other side. Moreover, the stochastic grouping method eliminated the oral hygiene

discrepancy from right handedness. Therefore, we could get a reliable result. However, split-mouth method also had disadvantage. High-pressure air and glycine particles could make small granules spread to the control side and ultrasonic scaling also has cavitation which have few influence on the other side. However, in consideration that the two sampling sites were far from each other, the impact could almost be neglected.

Mandibular first molars were chosen to be sampling sites. Because molars usually suffer more severe periodontitis comparing to incisor and premolar as a result of complex anatomical structure (*Jiao et al., 2017*). And cleaning first molars thoroughly is important and difficult for both patients and periodontists. If first molar could be effectively treated, other sites would get better therapeutic effect. In addition, the two investigational sites were far from each other that they would not be influenced by the intervention from the other side. Therefore, the clinical and microbiological effect of first molars of mandible were recorded and analyzed.

Ultrasonic scaling is the standard of care during maintenance therapy for its effectiveness and efficiency. Comparing with it, SGAP still has some advantages. Air polishing can be used on both natural teeth and restorative materials, while ultrasonic scaling is restricted to be used on restorative materials for its distorting effect (*Vermilyea, Prasanna & Agar, 1994*). SGAP is effective in removing stain and it is superior to ultrasonic scaling. Most of patients with periodontitis have exposed root during maintenance period for gingival recession. The application of ultrasonic scaling will leave a rough surface, and that is the reason why additional polishing with rubber-cup is needed; while the damage of glycine air polishing is much smaller. In addition, glycine air polishing can be used in patients who are afraid of the noise of ultrasonic scaling.

The safety of glycine air polishing has been reported in several research in vitro and in vivo (*Flemmig et al., 2012*; *Moene et al., 2010*; *Petersilka et al., 2002*), and our study confirmed the safety of SGAP again. Previous study had proved tooth stain and supragingival biofilm could be removed by supragingival air polishing (*Zhao, He & Meng, 2015*), and present study demonstrated subgingival periodontal pathogens could be removed as well. That is to say, supragingival air polishing could remove exogenous stain, supragingival and subgingival plaque of sites with pockets not more than 5 mm at the same time (*Flemmig et al., 2007*; *Patil et al., 2015*; *Petersilka et al., 2003c*, *2003d*), which makes it time-saving. The devices of supragingival air polishing and ultrasonic scaling are conventional used and every qualified periodontist and hygienist can operate them well. In addition, SPT with air polishing is more acceptable and easier to persist on for patients.

Considering the advantages and disadvantages of SGAP, it could be taken as a good alternative to remove biofilm of shallow pockets during maintenance period. And not more than 5 mm pockets during maintenance therapy without much calculus would be the indication for supragingival air polishing use. Further study is needed to compare the clinical and microbiological effect for longer term.

## CONCLUSION

Supragingival glycine air polishing had a reliable effect on periodontitis during maintenance therapy. It could remove soft deposits and improve clinical status, and the effect was similar to ultrasonic scaling and polishing with rubber-cup. Three months may be a proper maintenance interval for pockets not more than 5 mm.

## ACKNOWLEDGEMENTS

We especially thank Zhibin Chen, Xianghui Feng, Jian Jiao and Wenli Song (Department of Periodontology, Peking University School and Hospital of Stomatology, China) for their generous help.

### Funding

This work was supported by the National Science and Technology Pillar Program of the 11th Five-Year Plan of China (2007BAll8802) and the Project of the Key Clinical Disciplines of Ministry of Health of China (2010). The funders had no role in study design, data collection and analysis, decision to publish, or preparation of the manuscript.

### Grant Disclosures

The following grant information was disclosed by the authors:
National Science and Technology Pillar Program of the 11th Five-Year Plan of China: 2007BAll8802.
Project of the Key Clinical Disciplines of Ministry of Health of China (2010).

### Competing Interests

The authors declare that they have no competing interests.

### Author Contributions

- Hongye Lu conceived and designed the experiments, performed the experiments, analyzed the data, wrote the paper, prepared figures and/or tables.
- Lu He conceived and designed the experiments, contributed reagents/materials/analysis tools, clinical examination.
- Yibing Zhao conceived and designed the experiments, contributed reagents/materials/analysis tools, reviewed drafts of the paper.
- Huanxin Meng conceived and designed the experiments, contributed reagents/materials/analysis tools, reviewed drafts of the paper.

### Human Ethics

The following information was supplied relating to ethical approvals (i.e., approving body and any reference numbers):

The Peking University School and Hospital of Stomatology granted Ethical approval to carry out the study within its facilities: IRB00001052-05106.

## Clinical Trial Ethics

The following information was supplied relating to ethical approvals (i.e., approving body and any reference numbers):

This study was approved by Chinese Clinical Trial Registry.

## Clinical Trial Registration

The following information was supplied regarding Clinical Trial registration:

ChiCTR-INR-17013073.

## Data Availability

The raw data has been supplied as Supplemental Dataset Files.

## Supplemental Information

Supplemental information for this article can be found online at http://dx.doi.org/10.7717/peerj.4371#supplemental-information.

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
