# Peer review of "The effect of supragingival glycine air polishing on periodontitis during maintenance therapy: a randomized controlled trial"

_PeerJ, doi:10.7717/peerj.4371_

## Round 0.1 · original submission · Major Revisions

Please address the concerns raised by the reviewers for reconsideration in the journal for publication.

Reviewer 1 ·

Basic reporting

No comments

Experimental design

No comments

Validity of the findings

No comments

Additional comments

There is adequate literature on use of glycine air powder for non-surgical periodontal therapy and its comparison with ultrasonic debridement. The authors have reviewed some of it in relevant sections of the manuscript. It is standard of care to disrupt subgingival biofilm during periodontal maintenance and proposing to modify that well accepted treatment methodology will require strong scientific support. Perhaps for a convincing argument to be made in favor of supragingival debridement alone, a long-term comparison of clinical and microbiological parameters with subgingival debridement will be required. Glycine air powder is restricted in its ability to disrupt calculus deposits and an ultrasonic debridement for those areas would be standard of care. It is interesting to note that patients in this particular study did not have any calculus deposits during their maintenance phase (an assumption based on lack of mention of this in the manuscript).


Authors’ idea about patient comfort and time and cost-effective nature of supragingival plaque control that may have crossover effect on subgingival biofilm is the strongest hitherto largely unexplored question in this area. However, it is possible that the time and cost effectiveness of supragingival debridement alone would be diluted if cost of purchase, maintenance and training on both air-powder and ultrasonic instruments are taken in to account. No discussion was found on this aspect in the manuscript.

Overall, the manuscript needs significant revision from a grammatical and spelling standpoint. At multiple places throughout the manuscript, it was somewhat difficult to interpret statements.

Based on the overall deficiency in use of proper language and grammar, redundancy of already available data, development of clear rationale and lack of support for it restricts the utility of the manuscript in its current format.

Reviewer 2 ·

Basic reporting

I think that this manuscript would benefit from a review and thorough evaluation of English grammar and structure . There are spelling and grammar errors throughout the article, for example: Lines 72-73 (introduction) "And A.a. was less frequent than ultrasonic debridement group. Line 213 (results) "there was approximately significant difference between baseline and 12 weeks after treatment". Line 320 (discussion) I would replace "serious periodontitis" with "severe periodontitis".

Improper use of some references. For example: Loe's study (line 43) showed that dental plaque can cause gingivitis, not periodontitis.

Experimental design

Inadequate information on the randomization method. Please provide more details regarding the methods or software used to distribute the right or half of the mouth to the two study groups. Were all right quadrants assigned to the test group whereas the left ones to the control? If yes, can you please provide the rationale for this decision. As mentioned in the discussion, being left- or right-handed can have an effect on the performed oral hygiene. Were all patients right handed or left handed?

Other than that I think that is a well designed study.

Validity of the findings

I think the main limitation of the study is that it does not take into account the possible presence of calculus in the the periodontal patients and how this can affect the effectiveness of either of the two methods applied in this study. A main advantage of the ultrasonics, is the ability to remove calculus, and ability that air polishing is lacking as discussed in one of the references included (Moene et al. 2010). A 3-month period, is a long enough interval for plaque to become mineralized, opposite to what is stated in the discussion where White's paper is cited. There is a number of papers supporting that mineralization of plaque starts within a few days rather than months. The rate of calculus formation varies considerably between individuals.
Taking these factors into consideration, I think that sections of the paper should be revisited and as well as the conclusion (the recommendation for 3-month maintenance interval is based at an extend on the aforementioned parameters).

Additional comments

Overall I would say that the experimental design of the study was very good with the exception of the lack of details on the randomization methods.
Major revisions are recommended to address the issued discussed in the previous sections.
Thank you.

---

## Round 0.2 · accepted · Accept

Please correct the grammatical errors still present in your work (the instances mentioned by the reviewers are not comprehensive).

Congratulations on your work!

Reviewer 1 ·

Basic reporting

Okay

Experimental design

Okay

Validity of the findings

Okay

Additional comments

There are still minor grammatical errors in the manuscript.

Reviewer 2 ·

Basic reporting

There are minor grammar and spelling errors that could be addressed. For example:
Line 302: I would write "In the present study" instead of "In present study".
Line 341: "spilt-mouth" instead of "split-mouth".
Line 378: "remov1e" instead of "remove"

Experimental design

Well designed RCT, details provided on the randomization method, power analysis was performed.

Validity of the findings

Although clearly stated in the discussion, I would add in the conclusion section that the comparison is only valid for soft deposits and not calculus. When calculus is present, additional intervention is needed.

Additional comments

Dear authors,

Thank you for addressing the majority of the recommendations from the first review. I think at this point, there are only minor editing changes to be made. The manuscript describes a sound methodological approach and valid conclusions.

Thank you